# Closed-System Solution of the 1D Atom from Collision Model

**DOI:** 10.3390/e24020151

**Published:** 2022-01-19

**Authors:** Maria Maffei, Patrice A. Camati, Alexia Auffèves

**Affiliations:** CNRS, Grenoble INP, Institut Néel, Université Grenoble Alpes, 38000 Grenoble, France; patrice.camati@neel.cnrs.fr (P.A.C.); alexia.auffeves@neel.cnrs.fr (A.A.)

**Keywords:** collision model, quantum non-Markovian dynamics, input–output formalism, quantum optics, open quantum systems, repeated interaction model, quantum thermodynamics, waveguide quantum electrodynamics, quantum entanglement

## Abstract

Obtaining the total wavefunction evolution of interacting quantum systems provides access to important properties, such as entanglement, shedding light on fundamental aspects, e.g., quantum energetics and thermodynamics, and guiding towards possible application in the fields of quantum computation and communication. We consider a two-level atom (qubit) coupled to the continuum of travelling modes of a field confined in a one-dimensional chiral waveguide. Originally, we treated the light-matter ensemble as a closed, isolated system. We solve its dynamics using a collision model where individual temporal modes of the field locally interact with the qubit in a sequential fashion. This approach allows us to obtain the total wavefunction of the qubit-field system, at any time, when the field starts in a coherent or a single-photon state. Our method is general and can be applied to other initial field states.

## 1. Introduction

Since the establishment of quantum theory, quantum optics has described a plethora of phenomena of light-matter interactions coupling atomic degrees of freedom to few field modes, e.g., in cavity quantum electrodynamics (QED) [1] and more recently in circuit QED [2]. The characterization of the quantum state of the light-matter system, e.g., light-matter entanglement, is key to the experimental realization of information processes [3], quantum batteries [4], and to the fundamental research in quantum thermodynamics [5]. In past decades, an emerging endeavor arose, aiming to characterize the quantum properties of matter coupled to propagating field modes in waveguides, a field known as waveguide QED [6], that soon became central in the research on quantum computation [7], communication [8], and quantum thermodynamics [9,10,11,12,13]. In such systems, matter couples to an infinite continuum number of modes, rendering more difficult the characterization of light-matter entanglement from a full solution of the closed-system dynamics.

The paradigmatic setup of waveguide QED is a two-level atom (qubit) coupled to the field propagating in a one-dimensional waveguide, the so-called one-dimensional (1D) atom. This system can be experimentally implemented in several state-of-the-art platforms of integrated photonics [14], superconducting circuits [15,16], and atomic physics [17,18]. A number of approaches to obtain the solution of this system have been considered and here we mention but a few important ones. Notably, when the field and the qubit shares only one quantum of excitation, the Wigner–Weisskopf theory provides the total wavefunction solution [19,20]. For a few-photon or a coherent initial state, the closed-system solution has been solved in the Heisenberg picture, providing distribution functions of the output fields at any time [21], furthermore, the long-time limit of the field state has been derived from a scattering approach [22,23,24,25]. In addition, the joint state of the qubit and the output field can be obtained from a master equation coming from an effective model [26], capturing the entanglement between the atom and some degrees of freedom of the field. Our goal is to obtain the joint qubit-field state at any time employing the collision model (CM) framework that is naturally manifested within the 1D atom model as we now discuss it.

Collision models (CMs) have been widely employed to study the dynamics of open quantum systems [27]. They comprise a powerful and intuitive microscopic framework to derive Markovian and non-Markovian master equations [28,29]. The key underlying concept of CMs is to model the interaction between a system and an environment (bath) as a sequence of brief two-body “collisions” between the system and incoming bath units. The system state is computed at the end of each collision, leading to a stroboscopic evolution with discrete time. When the interaction time of each collision tends to zero and the number of bath units tends to infinite, one reaches the continuous-time limit and the master equations are obtained. In order to get a meaningful master equation from the CMs, it is usually assumed that the coupling constants between the system and the bath units diverge, a condition that may seem difficult to fulfil in nature. However, considering the 1D atom in the interaction picture with respect to the field Hamiltonian, and suitably defining discrete temporal mode operators, one can show that the CM framework is naturally manifested within the model [30,31], even containing the required diverging coupling. This result demonstrates waveguide QED as a suitable platform to physically realize the CM framework.

In the CM framework of the 1D atom, the electromagnetic field becomes the bath, regarded as an ensemble of discrete temporal modes that take the role of the bath units. The temporal modes, prepared in the input state, freely propagate in the waveguide until they reach the qubit position. At the qubit position, the temporal modes couple to the qubit one by one leading to a state change of both qubit and field. After the interaction, they keep propagating freely defining the output state, see Figure 1. As is usually the case for CMs, the master equation for the qubit can be obtained by tracing out the temporal modes after each collision. When the temporal modes of the field are initially uncorrelated, this leads to the well-known Optical Bloch Equation [32]. Moreover, from this CM view of the qubit-field interaction, one is also naturally led to an input–output view of the evolution, that is now built in the interaction picture instead of the usual input–output theory defined in the Heisenberg picture [33].

We apply the CM framework to two typical cases of quantum optics: a coherent input field and a single-photon wavepacket, highlighting a classical and a non-classical statistics, respectively. Considering an effective unitary operator for each collision between the qubit and the temporal modes, we are able to take the continuous-time limit of the joint qubit-field wavefunction at any time and hence obtain the solution of the dynamics.

The paper is organized as follows. In Section 2, we recall the collision model of the 1D atom firstly presented in Ref. [30]. In Section 3 and Section 4, we use this framework to derive the qubit-field wavefunctions for a coherent input field and for a single-photon input field, respectively.

## 2. Collision Model of the 1D Atom

We consider a qubit coupled with a multimode electromagnetic field in a one-dimensional chiral waveguide. The bare qubit Hamiltonian is given by $H_{q}=\hbar \omega_{q}\sigma_{+}\sigma_{-}$, with $\sigma_{-}=\left|g\rangle \langle e\right|$, $\sigma_{+}=\left|e\rangle \langle g\right|$, and $\sigma_{z}=\left|e\rangle \langle e\right|-\left|g\rangle \langle g\right|$, where |e〉 (resp. |g〉) denotes the qubit excited (resp. ground) state. The bare field Hamiltonian is given by $H_{f}=\sum_{k=0}^{\infty }\hbar \omega_{k}b_{k}^{†}b_{k}$, where $b_{k}^{†}$ (resp. $b_{k}$) creates (annihilates) one photon of discrete frequency $\omega_{k}=k\left(v2\pi /L\right)$, where we assumed that the field propagates from left to right with velocity *v* on a segment of length *L*, with periodic boundary condition. The operator $b_{k}$ satisfies the bosonic commutation relation $\left[b_{k},b_{k^{\prime }}^{†}\right]=\delta_{k,k^{\prime }}$. Furthermore, we assume that the field-qubit coupling is weak enough that only field frequencies near to $\omega_{q}$ are important (quasi-monochromatic approximation) [34], and that the coupling is uniform in frequency (first Markov approximation) [33]. In this case, the interaction Hamiltonian in the interaction picture with respect to $H_{q}+H_{f}$ reads $V\left(t\right)=i\hbar g\sum_{k}e^{i(\omega_{0}-\omega_{k})t}\sigma_{+}b_{k}+h.c.$, where h.c. stands for hermitian conjugate. Throughout the paper the operators in the interaction picture have their time dependence written explicitly.

In order to show how the CM framework arises naturally from the above Hamiltonian, we first rewrite the time-evolution operator as
(1)$$U\left(0,t\right)=\lim_{N\to \infty }\prod_{n=0}^{N-1}U\left(t_{n+1},t_{n}\right),$$
where $t_{n}=n\Delta t$, Δt=t/N, and
(2)$$U\left(t_{n},t_{n+1}\right)\equiv U_{n}=\exp\left\{-\frac{i}{\hbar }\Delta tV\left(t_{n}\right)\right\}.$$

In order to derive the last equality we have used the Magnus expansion [35] of the unitary evolution operator $U(t_{n},t_{n+1})$ at small Δt, where the terms featuring the commutator $\left[V\left(t^{\prime }\right),V\left(t^{\prime \prime }\right)\right]$ can be neglected as shown in [31]. The coupling Hamiltonian $V\left(t_{n}\right)$ can be also rewritten as
(3)$$V\left(t_{n}\right)=i\hbar \sqrt{\frac{\gamma }{\Delta t}}\left(\sigma_{+}\left(t_{n}\right)a_{n}-\sigma_{-}\left(t_{n}\right)a_{n}^{†}\right),$$
where $\gamma =g^{2}L/v$ and the temporal mode annihilation operator $a_{n}$ has been defined as
(4)$$a_{n}=\sqrt{\frac{v\Delta t}{L}}\sum_{k}e^{-i\omega_{k}t_{n}}b_{k}.$$

From the commutation relation for $b_{k}$, it follows that $\left[a_{n},a_{n}^{†}\right]=\delta_{n,n^{\prime }}$. Equations (Equation 1)–(Equation 3) imply that the evolution of the qubit-field state can be seen as a series of infinitesimal evolutions, i.e., collisions, that couple the qubit with only a single temporal mode $a_{n}$ at a time. For further reference, at time $t_{n}$ the state can be written as $\rho \left(t_{n}\right)=U_{n-1}\rho \left(t_{n-1}\right)U_{n-1}^{†}$.

The CM framework suggests intuitive definitions of input and output operators in the interaction picture. We define $a_{\mathrm{out}}\left(t_{n}\right)\equiv a_{n-1}/\sqrt{\Delta t}$, meaning that the output mode is the last temporal mode that has interacted with the qubit; we also define $a_{\mathrm{in}}\left(t_{n}\right)\equiv a_{n}/\sqrt{\Delta t}$, meaning that the input mode is the next temporal mode that is going to interact with the qubit, see Figure 1. Using these definitions, we can find a relation among the average values of input and output operators. We need to take $\Delta t\ll \gamma^{-1}$ so that we can approximate $U_{n}$ in Equation (Equation 2) to the second order in $V(t_{n})$, i.e., $U_{n}\approx 1-i\Delta tV\left(t_{n}\right)-\frac{V^{2}\left(t_{n}\right)}{2}\Delta t^{2}$, corresponding to the first order in Δt. Now, using Equation (Equation 3), we get the following expressions (at the first order in Δt):(5)$$\langle a_{\mathrm{out}}\left(t_{n}\right)\rangle =\mathrm{Tr}\left[\frac{U_{n-1}^{†}a_{n-1}U_{n-1}}{\sqrt{\Delta t}}\rho \left(t_{n-1}\right)\right]=\frac{\langle a_{n-1}\rangle }{\sqrt{\Delta t}}-\sqrt{\gamma }\langle \sigma_{-}\left(t_{n-1}\right)\rangle ;$$
and
(6)$$\langle a_{\mathrm{in}}\left(t_{n}\right)\rangle =\frac{\langle a_{n}\rangle }{\sqrt{\Delta t}}.$$

Putting together Equations (Equation 5) and (Equation 6), we get the discrete input–output relation
(7)$$\begin{array}{c} \langle a_{\mathrm{out}}\left(t_{n}\right)\rangle =\langle a_{\mathrm{in}}\left(t_{n-1}\right)\rangle -\sqrt{\gamma }\langle \sigma_{-}\left(t_{n-1}\right)\rangle . \end{array}$$

Let us notice that Equation (Equation 7) is the discrete-time analogue of the input–output relation which can be derived from the standard theory [33] in the Heisenberg picture, i.e., $\langle b_{\mathrm{out}}\left(t\right)\rangle =\langle b_{\mathrm{in}}\left(t\right)\rangle -\sqrt{\gamma }\langle \sigma_{-}\left(t\right)\rangle$.

## 3. Closed-System Solution for the Coherent Input Field

We now provide the qubit-field state solution when the field starts in a monochromatic coherent state $|\beta_{p}\rangle$ of frequency $\omega_{p}=\omega_{q}-\delta$. In order to comply with the quasi-monochromatic approximation, the detuning δ must satisfy $\left|\delta \right|\ll \omega_{q}$. This state is uncorrelated in the temporal domain and, therefore, under the interaction in Equation (Equation 3) gives rise to a Markovian qubit’s dynamics described by a Lindblad master equation [30].

The field’s state can be written as a product in the temporal mode basis:(8)$$\begin{array}{c} |\beta_{p}\rangle \equiv D\left(\beta_{p}\right)|0\rangle =\underset{n}{⨂}D\left(\alpha_{n}\right)|0_{n}\rangle \equiv \underset{n}{⨂}|\alpha_{n}\rangle , \end{array}$$
where $|0\rangle \equiv {⨂}_{n}|0_{n}\rangle$ is the field’s vacuum, $D\left(\beta_{p}\right)\equiv e^{\left(\beta_{p}b_{p}^{†}-\beta_{p}^{*}b_{p}\right)}$ is the displacement operator in the frequency domain, $D\left(\alpha_{n}\right)\equiv e^{\left(\alpha_{n}a_{n}^{†}-\alpha_{n}^{*}a_{n}\right)}$ is the one in the time domain, and $\alpha_{n}=\sqrt{v\Delta t/L}\beta_{p}e^{-i\omega_{p}t_{n}}$. To derive Equation (Equation 8) we plug the inverse of Equation (Equation 4) in the displacement operator $D(\beta_{p})$, obtaining $D\left(\beta_{p}\right)={⨂}_{n}D\left(\alpha_{n}\right)$, see also Ref. [30].

Provided that the qubit’s initial state is also a pure state $|\phi_{0}\rangle$, the system’s wavefunction at time $t_{N}$ is given by:(9)$$\begin{array}{cc} |\Psi (t_{N})\rangle = & \prod_{n=0}^{N-1}U_{n}|\beta_{p},\phi_{0}\rangle =D\left(\beta_{p}\right)D^{†}\left(\beta_{p}\right)\prod_{n=0}^{N-1}U_{n}D\left(\beta_{p}\right)|0,\phi_{0}\rangle =D\left(\beta_{p}\right)\underset{l}{⨂}D^{†}\left(\alpha_{l}\right)\left(\prod_{n=0}^{N-1}U_{n}\right)\underset{m}{⨂}D\left(\alpha_{m}\right)|0,\phi_{0}\rangle \\ = & D\left(\beta_{p}\right)\prod_{n=0}^{N-1}D^{†}\left(\alpha_{n}\right)U_{n}D\left(\alpha_{n}\right)|0,\phi_{0}\rangle \equiv D\left(\beta_{p}\right)\prod_{n=0}^{N-1}{\tilde{U}}_{n}|0,\phi_{0}\rangle . \end{array}$$

In the last equality we denoted by ${\tilde{U}}_{n}$ the collisional unitary operator in the displaced frame, which is defined as
(10)$$\begin{array}{c} {\tilde{U}}_{n}\equiv D^{†}\left(\alpha_{n}\right)U_{n}D\left(\alpha_{n}\right)=e^{-i\delta \Delta t\sigma_{+}\sigma_{-}+\frac{i\Omega \Delta t}{2}\sigma_{y}-\frac{i\Delta t}{\hbar }V\left(t_{n}\right)}, \end{array}$$
with $\Omega =2\sqrt{\gamma }\beta_{p}$. In the above expression, we also assumed that the frame rotates with the driving frequency, i.e., $\sigma_{-}\left(t_{n}\right)=e^{-i\omega_{p}t_{n}}\sigma_{-}$.

Due to the shape of the interaction, Equation (Equation 3), at each time $t_{n}$ at most one excitation of the field is absorbed or emitted. This implies that, when the field starts from the vacuum, its state at any time contains a maximum of one excitation for each temporal mode. It is clear that this condition simplifies the solution of the dynamics, for this reason, in the following, we will compute the wavefunction in the displaced frame, i.e., $|\tilde{\Psi }\left(t_{N}\right)\rangle =\prod_{n=0}^{N-1}{\tilde{U}}_{n}|0,\phi_{0}\rangle$. Let us notice that in order to come back to the the lab frame we just need to apply the operator $D(\beta_{p})$ to $|\tilde{\Psi }\left(t_{N}\right)\rangle$, see the last equality of Equation (Equation 9).

Once restricted the Fock basis of each temporal mode to $\{|0_{n}\rangle ,|1_{n}\rangle \}$, with $|1_{n}\rangle \equiv a_{n}^{†}|0_{n}\rangle$, the state $|\tilde{\Psi }\left(t_{N}\right)\rangle$ has the general shape
(11)$$|\tilde{\Psi }\left(t_{N}\right)\rangle =\sum_{\epsilon }\left[f_{\epsilon ,\phi_{0}}^{\left(0\right)}\left(t_{N}\right)+\sum_{n_{1}=0}^{N-1}f_{\epsilon ,\phi_{0}}^{\left(1\right)}\left(t_{N};t_{n_{1}}\right)a_{n_{1}}^{†}+\sum_{n_{1}=0}^{N-1}\sum_{n_{2}=n_{1}}^{N-1}f_{\epsilon ,\phi_{0}}^{\left(2\right)}\left(t_{N};t_{n_{1}},t_{n_{2}}\right)a_{n_{1}}^{†}a_{n_{2}}^{†}+\ldots \right]|0,\epsilon \rangle ,$$
where $\epsilon \in \left\{e,g\right\}$ denotes the qubit’s state, and the dots correspond to the components with m>2 photons emitted reading: $\sum_{n_{1}=0}^{N-1}\ldots \sum_{n_{m}=n_{m}-1}^{N-1}f_{\epsilon ,\phi_{0}}^{\left(m\right)}\left(t_{N};t_{n_{1}},\ldots ,t_{n_{m}}\right)a_{n_{1}}^{†}\ldots a_{n_{m}}^{†}|0,\epsilon \rangle$. In order to find the total wavefunction we need to find the explicit expression of the coefficients $f_{\epsilon ,\phi_{0}}^{\left(m\right)}\left(t_{N};t_{n_{1}},\ldots ,t_{n_{m}}\right)$.

Let us start with $f_{\epsilon ,\phi_{0}}^{\left(0\right)}\left(t_{N}\right)$, which is given by
(12)$$\begin{array}{c} f_{\epsilon ,\phi_{0}}^{\left(0\right)}\left(t_{N}\right)=\langle \epsilon ,0|\tilde{\Psi }\left(t_{N}\right)\rangle =\langle \epsilon |\left[\prod_{n=0}^{N-1}\langle 0_{n}|{\tilde{U}}_{n}|0_{n}\rangle \right]|\phi_{0}\rangle , \end{array}$$
where we used the fact that the operator ${\tilde{U}}_{n}$ only acts on the state of the qubit and of the *n*th temporal mode. In order to evaluate the qubit’s operator $\langle 0_{n}|{\tilde{U}}_{n}|0_{n}\rangle$, we expand the unitary evolution operator in Equation (Equation 10) up to the second order in $V(t_{n})$, corresponding to the first order in Δt, and we
(13)$$\begin{array}{c} \langle 0_{n}|{\tilde{U}}_{n}|0_{n}\rangle =1-i\delta \Delta t\sigma_{+}\sigma_{-}+\frac{i\Omega \Delta t}{2}\sigma_{y}-\frac{\gamma \Delta t}{2}\sigma_{+}\sigma_{-}\approx e^{-\gamma \Delta t/4-i\delta \Delta t/2}e^{\frac{-i\Delta t}{2}\left(\left(\delta -i\gamma /2\right)\sigma_{z}-\Omega \sigma_{y}\right)}, \end{array}$$
where the last equality becomes exact in the limit of $\Delta t\ll \gamma^{-1}$. Substituting the expression above in Equation (Equation 12), we find
(14)$$\begin{array}{c} f_{\epsilon ,\phi_{0}}^{\left(0\right)}\left(t_{N}\right)=e^{-\gamma t_{N}/4-i\delta t_{N}/2}\langle \epsilon |e^{\frac{-it_{N}}{2}\left[\left(\delta -i\gamma /2\right)\sigma_{z}-\Omega \sigma_{y}\right]}|\phi_{0}\rangle , \end{array}$$
which is straightforward to compute for any $|\phi_{0}\rangle$ and |ϵ〉. Performing this calculation for the four relevant terms, we obtain
(15)$$\begin{array}{cc} f_{g,g}^{\left(0\right)}\left(t\right) & =e^{-\gamma t/4-i\delta t/2}\left[\cos\left(\Omega^{\prime }t/2\right)+\sin\left(\Omega^{\prime }t/2\right)\left(\gamma +i2\delta \right)/\left(2\Omega^{\prime }\right)\right], \end{array}$$
(16)$$\begin{array}{cc} f_{g,e}^{\left(0\right)}\left(t\right) & =e^{-\gamma t/4-i\delta t/2}\sin\left(\Omega^{\prime }t/2\right)\Omega /\Omega^{\prime }, \end{array}$$
(17)$$\begin{array}{cc} f_{e,g}^{\left(0\right)}\left(t\right) & =-e^{-\gamma t/4-i\delta t/2}\sin\left(\Omega^{\prime }t/2\right)\Omega /\Omega^{\prime },\;\mathrm{and} \end{array}$$
(18)$$\begin{array}{cc} f_{e,e}^{\left(0\right)}\left(t\right) & =e^{-\gamma t/4-i\delta t/2}\left[\cos\left(\Omega^{\prime }t/2\right)-\sin\left(\Omega^{\prime }t/2\right)\left(\gamma +i2\delta \right)/\left(2\Omega^{\prime }\right)\right], \end{array}$$
where $\Omega^{\prime }=\sqrt{{\left(\Omega \right)}^{2}+{\left(\delta -i\gamma /2\right)}^{2}}$.

Now we can look at the component with one photon emitted:(19)$$f_{\epsilon ,\phi_{0}}^{\left(1\right)}\left(t_{N};t_{n_{1}}\right)=\langle \epsilon ,0|a_{n_{1}}|\tilde{\Psi }\left(t_{N}\right)\rangle =\langle \epsilon |\left[\prod_{n=n_{1}+1}^{N-1}\langle 0_{n}|{\tilde{U}}_{n}|0_{n}\rangle \right]\langle 0_{n_{1}}|a_{n_{1}}{\tilde{U}}_{n_{1}}|0_{n_{1}}\rangle \left[\prod_{n=0}^{n_{1}-1}\langle 0_{n}|{\tilde{U}}_{n}|0_{n}\rangle \right]|\phi_{0}\rangle .$$

Here we just need to evaluate the qubit’s operator $\langle 0_{n}|a_{n}{\tilde{U}}_{n}|0_{n}\rangle$, which gives (at the first order in Δt)
(20)$$\begin{array}{c} \langle 0_{n}|a_{n}{\tilde{U}}_{n}|0_{n}\rangle =-\sqrt{\gamma \Delta t}\sigma_{-}\left(t_{n}\right)=-\sqrt{\gamma \Delta t}\sigma_{-}e^{-i\omega_{p}t_{n}}. \end{array}$$

Plugging the expression above in Equation (Equation 19) we get
(21)$$\begin{array}{c} f_{\epsilon ,\phi_{0}}^{\left(1\right)}\left(t_{N};t_{n_{1}}\right)=-\sqrt{\gamma \Delta t}f_{\epsilon ,g}^{\left(0\right)}\left(t_{N}-t_{n_{1}}\right)e^{-i\omega_{p}t_{n_{1}}}f_{e,\phi_{0}}^{\left(0\right)}\left(t_{n_{1}}\right). \end{array}$$

Repeating the same strategy we can obtain $f_{\epsilon ,\phi 0}^{\left(2\right)}\left(t_{N};t_{n_{1}},t_{n_{2}}\right)$:(22)$$\begin{array}{cc} f_{\epsilon ,\phi_{0}}^{\left(2\right)}\left(t_{N};t_{n_{1}},t_{n_{2}}\right)=\langle \epsilon ,0|a_{n_{1}}a_{n_{2}}|\tilde{\Psi }\left(t_{N}\right)\rangle = & \langle \epsilon |\left[\prod_{n=n_{2}+1}^{N-1}\langle 0_{n}|{\tilde{U}}_{n}|0_{n}\rangle \right]\langle 0_{n_{2}}|a_{n_{2}}{\tilde{U}}_{n_{2}}|0_{n_{2}}\rangle \left[\prod_{n=n_{1}+1}^{n_{2}-1}\langle 0_{n}|{\tilde{U}}_{n}|0_{n}\rangle \right]\\ \; & \times \langle 0_{n_{1}}|a_{n_{1}}{\tilde{U}}_{n_{1}}|0_{n_{1}}\rangle \left[\prod_{n=0}^{n_{1}-1}\langle 0_{n}|{\tilde{U}}_{n}|0_{n}\rangle \right]|\phi_{0}\rangle . \end{array}$$

Evaluating all the terms, we find
(23)$$\begin{array}{c} f_{\epsilon ,\phi_{0}}^{\left(2\right)}\left(t_{N};t_{n_{1}},t_{n_{2}}\right)=\gamma \Delta tf_{\epsilon ,g}^{\left(0\right)}\left(t_{N}-t_{n_{2}}\right)e^{-i\omega_{p}t_{n_{2}}}f_{e,g}^{\left(0\right)}\left(t_{n_{2}}-t_{n_{1}}\right)e^{-i\omega_{p}t_{n_{1}}}f_{e,\phi_{0}}^{\left(0\right)}\left(t_{n_{1}}\right). \end{array}$$

From these and the other terms not shown, we find the explicit expression of the functions $f_{\epsilon ,\phi_{0}}^{\left(m\right)}\left(t_{N};t_{n_{1}},\ldots ,t_{n_{m}}\right)$ for any number m≥2 of photons emitted as
(24)$$f_{\epsilon ,\phi_{0}}^{\left(m\right)}\left(t_{N};t_{n_{1}},\ldots ,t_{n_{m}}\right)={\left(-\sqrt{\gamma \Delta t}\right)}^{m}f_{\epsilon ,g}^{\left(0\right)}\left(t_{N}-t_{n_{m}}\right)e^{-i\omega_{p}t_{n_{m}}}\left[\prod_{i=2}^{m}f_{e,g}^{\left(0\right)}\left(t_{n_{i}}-t_{n_{i-1}}\right)e^{-i\omega_{p}t_{n_{i-1}}}\right]f_{e,\phi_{0}}^{\left(0\right)}\left(t_{n_{1}}\right).$$

The continuous-time version of Equation (Equation 11) can be found taking the limits $t_{N}\to t$, $a_{n}^{†}/\sqrt{\Delta t}\to a^{†}\left(t\right)$, and $\sum_{n=0}^{N-1}\Delta t\to \int_{0}^{t}dt^{\prime }$.

When the input field is intense, namely γ≪Ω, and resonant with the qubit, the expressions of the coefficients simplify since we can take $\Omega^{\prime }\approx \Omega$ and γ/Ω≈0. Furthermore, we can neglect the terms containing multiple photon emissions, whose probability goes to zero. In this case, we find the simple expression
(25)$$\begin{array}{cc} |\tilde{\Psi }\left(t\right)\rangle \approx & e^{-\gamma t/4}\left[\cos\left(\frac{\Omega t}{2}\right)-\sqrt{\gamma }\int_{0}^{t}dt^{\prime }\cos\left(\frac{\Omega (t-t^{\prime })}{2}\right)\sin\left(\frac{\Omega t^{\prime }}{2}\right)e^{-i\omega_{q}t^{\prime }}a^{†}\left(t^{\prime }\right)\right]|0,g\rangle \\ \; & +e^{-\gamma t/4}\left[\sin\left(\frac{\Omega t}{2}\right)-\sqrt{\gamma }\int_{0}^{t}dt^{\prime }\sin\left(\frac{\Omega (t-t^{\prime })}{2}\right)\sin\left(\frac{\Omega t^{\prime }}{2}\right)e^{-i\omega_{q}t^{\prime }}a^{†}\left(t^{\prime }\right)\right]|0,e\rangle , \end{array}$$
where we set the qubit’s initial state to the ground state. We note that in the long-time limit, t→∞, the field state obtained from our solution is consistent with the one derived in Ref. [25] using a generalization of scattering theory.

Finally, our method provides the well-known Wigner–Weisskopf solution [19] when the initial field’s state is |0〉 and the initial qubit’s state is |e〉. In this case, due to the conservation of the total number of excitations, we expect a solution of the form
(26)$$\begin{array}{c} |\Psi \left(t_{n}\right)\rangle =h_{e,e}^{\left(0\right)}\left(t_{N}\right)|0,e\rangle +\sum_{n_{1}=0}^{N-1}h_{g,e}^{\left(1\right)}\left(t_{N};t_{n_{1}}\right)a_{n_{1}}^{†}|0,g\rangle , \end{array}$$
with:(27)$$\begin{array}{cc} h_{e,e}^{\left(0\right)}\left(t_{N}\right)= & \langle e,0|\Psi \left(t_{N}\right)\rangle =\langle e|\left[\prod_{n=0}^{N-1}\langle 0_{n}|U_{n}|0_{n}\rangle \right]|e\rangle =\langle e|e^{-\gamma t_{N}\sigma_{+}\sigma_{-}/2}|e\rangle =e^{-\gamma t_{N}/2}, \end{array}$$
(28)$$\begin{array}{cc} h_{g,e}^{\left(1\right)}\left(t_{N};t_{n_{1}}\right)= & \langle g,0|a_{n}\Psi \left(t_{N}\right)\rangle =\langle g|\left[\prod_{n=n_{1}+1}^{N-1}\langle 0_{n}|U_{n}|0_{n}\rangle \right]\langle 0_{n_{1}}|a_{n_{1}}U_{n_{1}}|0_{n_{1}}\rangle \left[\prod_{n=0}^{n_{1}-1}\langle 0_{n}|U_{n}|0_{n}\rangle \right]|e\rangle \\ = & -\sqrt{\gamma \Delta t}e^{-i\omega_{q}t_{n_{1}}}h_{e,e}^{\left(0\right)}\left(t_{n_{1}}\right)=-\sqrt{\gamma \Delta t}e^{-\left(\gamma +i\omega_{q}\right)t_{n_{1}}}. \end{array}$$

Plugging Equation (Equation 27) in Equation (Equation 26) and taking the continuous-time limit we get
(29)$$\begin{array}{c} |\Psi \left(t\right)\rangle =e^{-\gamma t/2}|0,e\rangle -\sqrt{\gamma }\int_{0}^{t}dt^{\prime }e^{-\gamma t^{\prime }/2-i\omega_{q}t^{\prime }}a^{†}\left(t^{\prime }\right)|0,g\rangle . \end{array}$$

## 4. Closed-System Solution for the Single-Photon Input Field

Here we provide the solution when the field starts in a single-photon wavepacket of central frequency $\omega_{p}=\omega_{q}-\delta$ with $\delta \ll \omega_{q}$:(30)$$\begin{array}{c} |1_{p}\rangle =\sum_{n=0}^{\infty }\sqrt{\Delta t}\xi \left(t_{n}\right)a_{n}^{†}|0\rangle , \end{array}$$
with $\sum_{n=0}^{\infty }\Delta t{\left|\xi \left(t_{n}\right)\right|}^{2}=1$. The field in Equation (Equation 30) is already correlated in the temporal domain before even interacting with the qubit, i.e., it can not be written as a product state of the individual temporal modes: the resulting qubit’s reduced dynamics is non-Markovian [36,37]. In order to solve the full dynamics, we use a more general strategy than the one used in the previous section. We replace $U_{n}$, given by Equation (Equation 2), with an effective unitary map $M_{n}$ having an equivalent action in the limit of $\Delta t\ll \gamma^{-1}$. Applying this map repeatedly to the initial state leads to recursive relations for the unnormalized field states $|\psi_{e}\left(t_{N}\right)\rangle$ and $|\psi_{g}\left(t_{N}\right)\rangle$ contained in the joint qubit-field wavefunction:(31)$$\begin{array}{c} |\Psi \left(t_{N}\right)\rangle =U_{N-1}\ldots U_{0}|\Psi \left(t_{0}\right)\rangle \approx M_{N-1}\ldots M_{0}|\Psi \left(t_{0}\right)\rangle \equiv |\psi_{e}\left(t_{N}\right),e\rangle +|\psi_{g}\left(t_{N}\right),g\rangle . \end{array}$$

This effective map reads
(32)$$\begin{array}{cc} \; & M_{n}|g,\psi_{g}\left(t_{n}\right)\rangle =e^{\frac{-\gamma \Delta t}{2}}|g,\psi_{g}\left(t_{n}\right)\rangle +\sqrt{1-e^{-\gamma \Delta t}}e^{i\omega_{q}t_{n}}a_{n}|e,\psi_{g}\left(t_{n}\right)\rangle , \end{array}$$
(33)$$\begin{array}{cc} \; & M_{n}|e,\psi_{e}\left(t_{n}\right)\rangle =e^{\frac{-\gamma \Delta t}{2}}|e,\psi_{e}\left(t_{n}\right)\rangle -\sqrt{1-e^{-\gamma \Delta t}}e^{-i\omega_{q}t_{n}}a_{n}^{†}|g,\psi_{e}\left(t_{n}\right)\rangle . \end{array}$$

The effective map $M_{n}$ gives the following recursive relations:(34)$$\begin{array}{cc} \; & |\psi_{g}\left(t_{n}\right)\rangle =e^{-\gamma \Delta t/2}|\psi_{g}\left(t_{n-1}\right)\rangle -\sqrt{1-e^{-\gamma \Delta t}}e^{-i\omega_{q}t_{n}}a_{n}^{†}|\psi_{e}\left(t_{n-1}\right)\rangle ,\;\mathrm{and} \end{array}$$(35)$$\begin{array}{cc} \; & |\psi_{e}\left(t_{n}\right)\rangle =e^{-\gamma \Delta t/2}|\psi_{e}\left(t_{n-1}\right)\rangle +\sqrt{1-e^{-\gamma \Delta t}}e^{i\omega_{q}t_{n}}a_{n}|\psi_{g}\left(t_{n-1}\right)\rangle . \end{array}$$

When the initial state of the qubit is the ground state, i.e., $|\Psi \left(t_{0}\right)\rangle =|g,1_{p}\rangle$, the expressions above give rise to a closed form:
(36)$$\begin{array}{cc} |\psi_{g}\left(t_{N}\right)\rangle = & \sum_{n=0}^{N-1}\left[\sqrt{\Delta t}\xi \left(t_{n}\right)-\gamma \Delta te^{-\frac{\gamma t_{n}}{2}-i\omega_{q}t_{n}}\sum_{m=0}^{n}\left(e^{\frac{\gamma t_{m}}{2}+i\omega_{0}t_{m}}\sqrt{\Delta t}\xi \left(t_{m}\right)\right)\right]a_{n}^{†}|0\rangle \\ & +\sum_{n=N}^{\infty }\sqrt{\Delta t}\xi \left(t_{n}\right)a_{n}^{†}|0\rangle , \end{array}$$(37)$$\begin{array}{cc} |\psi_{e}\left(t_{N}\right)\rangle = & \sqrt{\gamma \Delta t}e^{-\frac{\gamma t_{N}}{2}}\sum_{n=0}^{N-1}e^{\frac{\gamma t_{n}}{2}+i\omega_{q}t_{n}}\sqrt{\Delta t}\xi \left(t_{n}\right)|0\rangle . \end{array}$$

Substituting the equations above in Equation (Equation 31) and taking the continuous-time limit, we get
(38)$$\begin{array}{c} |\Psi \left(t\right)\rangle =\sqrt{\gamma }\tilde{\xi }\left(t\right)|0,e\rangle +\left(\int_{t}^{\infty }dt^{\prime }\xi \left(t^{\prime }\right)\right.+\left.\int_{0}^{t}dt^{\prime }\left[\xi \left(t^{\prime }\right)-\gamma \tilde{\xi }\left(t^{\prime }\right)e^{-i\omega_{q}t^{\prime }}\right]\right)a^{†}\left(t^{\prime }\right)|0,g\rangle , \end{array}$$
with $\tilde{\xi }\left(t\right)=e^{-\gamma t/2}\int_{0}^{t}dt^{\prime }\left[e^{\frac{\gamma t^{\prime }}{2}+i\omega_{q}t^{\prime }}\xi \left(t^{\prime }\right)\right]$. Let us notice that tracing Equation (Equation 38) over the field, we obtain the qubit’s state derived in Refs. [36,37], while taking its long-time limit we obtain the final field’s state derived in Ref. [22].

Finally, we point out that the effective map in Equation (Equation 32) can, in principle, be used to derive the full system’s wavefunction with any kind of input field. According on the input state, getting a closed form for the total wavefunction may be more or less complicated. In the spontaneous emission case, for example, it is straightforward to verify that the effective map provides the expected solution given by Equation (Equation 29).

## 5. Conclusions

We derived the analytical solution of the 1D atom’s closed-dynamics using the CM framework. We analyzed two paradigmatic cases corresponding to different input fields, i.e., a coherent and a single-photon field. These fields give rise, respectively, to a Markovian and a non-Markovian qubit’s reduced dynamics. We showed that, besides being useful to derive master equations, CMs are also useful to derive qubit-field wavefunctions. The method presented is general and can be applied to other kinds of input fields or to different shapes of the qubit-field interaction allowing for a CM treatment. Framing the CM into a closed-system approach can shed new light on the thermodynamical analysis. Indeed, the pure state of the full system provides access to the correlations between the qubit and the bath, and among different time units of the bath, possibly revealing the microscopic mechanism underlying the total entropy production [38].

## Figures and Tables

**Figure 1 entropy-24-00151-f001:**
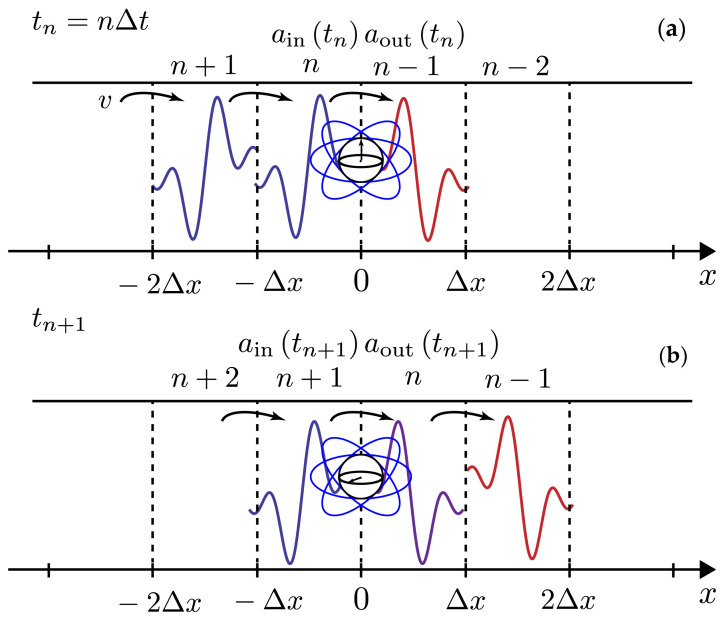
Collision model of the 1D atom. The field, propagating from left to right with constant velocity *v*, is decomposed into discrete temporal modes created by the bosonic operators $a_{n}^{†}$, see Equation (Equation 4). Time and space are considered to be discrete, i.e., $t\to t_{n}=n\Delta t$ and $x\to x_{n}=nv\Delta t$. (**a**) Snapshot of the system at time $t_{n}$, beginning of the *n*th collision. The temporal mode created by the operator $a_{n}^{†}$ is arriving at the qubit position (x=0) where it is going to interact, then it defines the input operator, i.e $a_{\mathrm{in}}\left(t_{n}\right)=a_{n}/\sqrt{\Delta t}$. The temporal mode created by the operator $a_{n-1}^{†}$ that just interacted with the qubit defines the output operator $a_{\mathrm{out}}\left(t_{n}\right)=a_{n-1}/\sqrt{\Delta t}$. (**b**) Snapshot of the system at time $t_{n+1}$, beginning of the (n+1)th collision. Now, the mode number n+1 defines the input, and the mode number *n* defines the output. The state of the qubit changed with respect to the time $t_{n}$ due to the past collision with the mode number *n*.

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
