# Peer review of "Closed-System Solution of the 1D Atom from Collision Model"

_entropy, 2022, doi:10.3390/e24020151_

Round 1

Reviewer 1 Report

The authors exploit collision models (CMs) to solve the closed dynamics of a system composed of a single atom coupled to a a field flowing through a one-dimensional waveguide (1D atom). They study the cases in which the field is assumed to be in a coherent state and in single-photon pulse.
The manuscript is well written and shows very clearly how CMs can be useful to describe the overall quantum dynamics of atoms-field systems through closed-system (unitary) formalism, as well as quantum dynamical maps (open system approach).
Part of the paper (in particular section 3) features many technicalities but in my opinion this does not affect the readability.
Therefore I recommend the manuscript for publication in Entropy.

I have just two comments:

1) Probably in line 143 (section 3) it should be D^\dagger

2) Just for the sake of clarity, can the authors better explain why they have chosen to introduce non-normalised states in Eq (31) and the map M in section 4?

Reviewer 2 Report

I like this paper and I recommend it for publication. It discusses a timely topic, which connects collision models with realistic quantum optical setups. In the first part, the authors provide a very nice pedagogical introduction to the mapping between a waveguide QED model and a collision model. Then they move on to compute the full solution for two particular initial conditions of the field. I have only minor comments.

  • The authors take only k > 0. I understand that this is meant to reflect the fact that the pulse is propagating from left to right. But in principle isn't it necessary to also include the modes with k < 0? Could the authors please comment on this?
  • Around Eq. (3) and (4): could the authors perhaps clarify a bit better what are the crucial approximations/hypotheses that convert the original model to a CM? Is it the assumption that [V(t'), V(t'')] \simeq 0?
  • In my opinion, this issue of the approximations is a big problem if one is toprovide a solid link between waveguide QED and CMs. I would recommend to the authors to summarize, perhaps in the conclusions, what kinds of approximations/assumptions were necessary throughout the paper. For example, in Sec. III the authors assume the detuning \delta is small. How crucial is this? And what happens if this is not the case? It would be nice to have an overview of all approximations employed. And also to know which ones are necessary, and which are introduced for convenience.  
